# Three Phenolic Extracts Regulate the Physicochemical Properties and Microbial Community of Refrigerated Channel Catfish Fillets during Storage

**DOI:** 10.3390/foods12040765

**Published:** 2023-02-09

**Authors:** Jian Huang, Lan Wang, Zhenzhou Zhu, Yun Zhang, Guangquan Xiong, Shuyi Li

**Affiliations:** 1National R&D Center for Se-Rich Agricultural Products Processing, College of Food Science and Engineering, Wuhan Polytechnic University, Wuhan 430023, China; 2Key Laboratory of Cold Chain Logistics Technology for Agro-Product, Institute of Agricultural Products Processing and Nuclear-Agricultural Technology, Hubei Academy of Agricultural Sciences, Ministry of Agriculture and Rural Affairs, Wuhan 430064, China; 3Hubei Engineering Research Center for Deep Processing of Green Se-Rich Agricultural Products, School of Modern Industry for Selenium Science and Engineering, Wuhan Polytechnic University, Wuhan 430023, China; 4College of Tourism and Hotel Management, Hubei University of Economics, Wuhan 430205, China

**Keywords:** polyphenol, catfish fillets, physicochemical changes, microbial community, storage

## Abstract

It has been demonstrated that polyphenols have the potential to extend the shelf life of fish products. Thus, the effects of phenolic extracts from grape seed (GSE), lotus seedpod (LSPC), and lotus root (LRPE) were investigated in this study, focusing on the physicochemical changes and bacterial community of refrigerated channel catfish fillets during storage at 4 °C, using ascorbic acid (AA) as reference. As a result, GSE, LSPC, LRPE and AA inhibit the reproduction of microbials in catfish fillets during storage. According to the microbial community analysis, the addition of polyphenols significantly reduced the relative abundance of *Proteobacterial* in the early stage and changed the distribution of the microbial community in the later stage of storage. After 11 days of storage, the increase in total volatile base nitrogen (TVB-N) in fish was significantly reduced by 25.85%, 25.70%, 22.41%, and 39.31% in the GSE, LSPC, LRPE, and AA groups, respectively, compared to the control group (CK). Moreover, the lipid oxidation of samples was suppressed, in which thiobarbituric acid-reactive substances (TBARS) decreased by 28.77% in the GSE group, compared with the CK. The centrifugal loss, LF-NMR, and MRI results proved that GSE significantly delayed the loss of water and the increase in immobilized water flowability in catfish fillets. The polyphenol-treated samples also showed less decrease in shear force and muscle fiber damage in histology, compared to the CK. Therefore, the dietary polyphenols including GSE, LSPC, and LRPE could be developed as natural antioxidants to protect the quality and to extend the shelf life of freshwater fish.

## 1. Introduction

Native to Canada and Mexico, the channel catfish (*Ictalurus puctactus*) is a major aquaculture species in the United States. Since the introduction of channel catfish in China in 1984, the aquaculture scale has been expanding, and the market potential has been gradually explored [1]. From 2016, China has become the main channel catfish producer around the world with an annual yield of channel catfish reaching 280,000 tons [2]. Even though channel catfish lack intramuscular spines and scales, they are abundant in vitamins, unsaturated fatty acids, and high-quality protein [3]. As the main processing products, the refrigerated channel catfish fillets are easy to cook and store, which makes them popular with consumers [4]. However, the storage and transportation of refrigerated catfish fillets are limited due to their susceptibility to bacterial contamination and decay during storage, which greatly reduces their economic benefits [3]. Consequently, it is crucial to develop an effective preservation stratagem that can delay the quality decline of catfish fillets and prolong the shelf life of catfish fillets.

In recent decades, plant polyphenols have been widely used in the preservation of aquatic products because of their excellent antioxidative and bacteriostatic efficacy [5]. Grape seed extract (GSE), as a typical natural plant extract, contains abundant polyphenols, such as catechin, epicatechin, and proanthocyanidins. It not only has excellent antioxidant activities by scavenging free radicals, but it also has antibacterial properties due to the core structures containing 3,4,5-trihydroxyphenyl groups [6]. Shi, et al. [7], for example, discovered that grape seed extracts can prevent lipid and protein oxidation and extend the shelf life of silver carp fillets in chilled storage. Lotus seedpod proanthocyanidins (LSPC) are complex polyphenols composed of flavan-3-ol units. It has been reported that LSPC not only has high safety and antioxidant activity (including scavenging superoxide anion inhibiting the formation of malondialdehyde and advanced glycation end products), but it also has strong antibacterial activity [8,9,10,11]. Li, Wang, Gao, Xie and Sun [8] investigated the effects of lotus seedpod procyanidins on the quality of chilled beef during refrigerated storage and observed that LSPC could not only inhibit lipid oxidation but could also delay the denaturation of proteins. The main components of lotus root polyphenol extract (LRPE) include chlorogenic acid, B-type procyanidin dimer·H_2_O, (+)-Catechin, (−)-Epicatechin, propyl gallate·H_2_O, caffeic acid, (−)-Epicatechin-3-gallate, and rutin [12], which has also exhibited high antioxidative capacity [13].

On an account of the abundant available nutrients, high water content and neutral pH, catfish fillets became the suitable substrate for microorganism growth and reproduction. According to statistics, about 30% of economic losses were caused by microorganisms [3,14]. Even though the mechanism of how polyphenols kill or inhibit bacteria is still unclear, studies have shown that the interactions between polyphenols and nonspecific force, including hydrogen-bonding, hydrophobic interactions, and the formation of covalent bonds, were related to microbial membranes, enzymes, and cell envelope transport proteins [15,16]. The polyphenols’ capacity to interact with bacterial cell wall components and the bacterial cell membrane allows them to inhibit and control biofilm formation, as well as inhibit microbial enzymes, interfere with protein regulation, and deprive bacterial cell enzymes of substrates [17]. Meanwhile, polyphenols have the capacity to chelate metal ions, vital for survival of almost all bacteria, which may be an important reason for their antibacterial properties [17,18]. Therefore, polyphenols are effective antimicrobials. 

Previous research has demonstrated the composition and antioxidant activity of grape seed extract (GSE), lotus seedpod procyanidins (LSPC) and lotus root polyphenol extract (LRPE), all of which have high potential as preservatives. However, there is no study to confirm and compare their fresh-keeping capacity. As a consequence, the overall objective of the present study was to identify how the addition of GSE, LSPC, and LRPE, as well as ascorbic acid (AA) can retard both lipid and protein oxidation, inhibit the growth of meat spoilage bacteria and extend the shelf life of catfish fillets. Moreover, this study provided greater insight into the potential of lotus root, grape seed, and lotus seedpod natural extract as natural and effective sources of antioxidants and antibacterials for fish processing.

## 2. Materials and Methods

### 2.1. Preparation of Samples 

The lotus root polyphenol extract (LRPE) was extracted from rhizome knots according to Zhu, Li, He, Thirumdas, Montesano and Barba [12]. Following that, the crude extract solution was loaded onto the AB-8 macroporous resin (0.3–1.25 mm particle size, Macklin, Shanghai, China) for purification as described by Wu, et al. [19]. The rhizome knot was purchased from a local market (Wuhan, China). The concentration of total polyphenol in LRPE determined by the method of Zhu, Li, He, Thirumdas, Montesano and Barba [12] was 68.73 mg/100 mg. The ascorbic acid (AA) (99.7%) was purchased from Sinopharm Chemical Reagent Co., Ltd. (Shanghai, China). The lotus seedpod procyanidins (LSPC) (50%) were provided by College of Food Science and Technology, Huazhong Agricultural University. The grape seed extract (GSE) (95%) was obtained from Tianjin Jianfeng Natural Products Research and Development Co., Ltd. (Tianjin, China).

Fresh and alive channel catfish, each 3 ± 0.2 kg, were purchased from a local market (Wuhan, China). All channel catfish were transported using water tanks to the laboratory with 30 min. The fish head, bone, viscera and skin were immediately removed. The back muscles of the fish were cut into approximately 2 cm thick and approximately 20 g fillets and then randomly separated into five groups: a CK group treated with sterile distilled water and four other treatment groups including AA, GSE, LSPC and LRPE at a concentration of 2 g/L. The concentration of each treatment group was determined according to the results of previous studies. Each solution was prepared just before use and precooled to 4 °C. Fillets were immersed in each treatment group for 10 min at room temperature before being drained thoroughly. The fillets were then placed in food packaging boxes at a cold storage temperature (4 ℃) and were collected at random for analysis at 0, 1, 3, 5, 7, 9, and 11 days.

### 2.2. Total Viable Counts (TVC)

The TVC of the fish meat sample was determined by using the method of plate counting described by Zhang, et al. [20] with some modifications. Briefly, samples (1 mL) of serial dilutions (1: 9, sterile 0.85% saline) of homogenates were mixed with the plate count agar (PCA). The plate was cultured upside down at 30 °C for 72 h after the agar solidified. The microbial counts were expressed as log CFU/g.

### 2.3. High-Throughput Sequencing on the Illumina Platform

Extraction of bacterial DNA followed the method described by Li, et al. [21] with some modifications. Homogenates (10 mL, obtained in Section 2.2) were centrifuged at 13,000× *g* for 10 min. After that, the supernatants were removed and sediments were washed by 1 mL sterile water. Bacterial DNA was extracted from the sediments with the assay kit of bacterial DNA extraction (Biomed Biological Technology Co., Ltd., Beijing, China). The V3 and V4 variable regions of bacterial 16S rDNA gene were amplified with the 338F (5′-ACTCCTACGGGAGGCAGCA-3′) and 806R (5′-GGACTACHVGGGTWTCTAA-3′) primers with different barcodes. The TruSeq Nano DNA LT library preparation kit (Illumina Inc., San Diego, CA, USA) was used to produce sequencing library. Operational taxonomic units (OTU) were picked only if they had similarity values of 97% or higher. R software (version 2.15.3, The University of Auckland, Auckland, New Zealand) was used to create a heat map based on the relative abundance of microbial genera.

### 2.4. Total Volatile Base Nitrogen (TVB-N)

The TVB-N of the fish meat sample was determined by using an Automatic Kjeldahl Nitrogen Determinator (K9860, Hai Neng Instrument Co., Ltd., Shanghai, China), according to the method described by Sun, et al. [22] with some modifications. Briefly, 5 g of minced fish meat was mixed with 75 mL distilled water and soaked for 30 min after full vibration at 4 °C. The mixture was then filtered. Then, 5 mL of filtrate was mixed with 5 mL of 10 g/L MgO solution and was distilled for 6 min with kjeldahl. The receiving solution was titrated with 0.01 mol/L hydrochloric acid standard titration solution, with blue-purple as the end point of titration. All chemicals were provided by sigma (Steinheim, Germany). 

### 2.5. Thiobarbituric Acid Reactive Substance (TBARS)

The TBARS value was determined according to the method described by Sun, et al. [23]. Briefly, 2 g of minced fish meat was mixed with 200 μL butyl hydroxyanisole (BHA, 7.2%, *w*/*v* 98% ethanol) and 7 mL trichloroacetic acid (TCA, 5%). Filter after homogenization. The filtrate was then diluted ten times, and 5 mL of diluent was mixed with 5 mL 0.2 M TBA, followed by heated at 80 °C for 60 min. The absorbance at 532 nm was recorded after cooling.

### 2.6. Shear Force

The shear force of samples was determined according to the method described by Pinheiro, et al. [24]. Briefly, five cores in cubes shape (2.0 × 3.0 × 3.0 cm) vertical to the fiber direction were obtained. The cores were then sheared to a cross-section of the muscle fiber by a Warnerؘ–Bratzer shear blade, using a TA-XT Plus Texure Analyzer (Stable Micro System Ltd., UK). The parameters in shear force (g) measurement were set as follows: testing speed = 1.0 mm/s, distance = 25 mm, and trigger force = 5 g.

### 2.7. Histology 

The histology of catfish samples was preformed according to the method described by Shao, et al. [25]. The samples were fixed, dehydrated, and embedded in paraffin. After hematoxylin–eosin staining, the microstructure of the tissue sections was observed using an optical microscope (Eclipse Ci, Nikon, Tokyo, Japan).

### 2.8. Centrifugal Loss

The centrifugal loss was measured according to the method described by Li, et al. [21] with some modifications. Then, 2 g of minced sample wrapped by absorbent paper was put into a 50 mL centrifuge tube and centrifuged at 4000 rpm using a refrigerated centrifuge (CR21N, Hitachi, Japan) at 4 °C for 10 min. Subsequently, the sample was put out and weighed. 

### 2.9. Low-Field Nuclear Magnetic Resonance (LF-NMR) and Magnetic Resonance Imaging (MRI)

Low-field nuclear magnetic resonance (LF-NMR) relaxation measurement and magnetic resonance imaging (MRI) analysis were preformed according to the method described by Xia, et al. [26] using an NMR analyzer (NIUMAG, Shanghai, China). Briefly, the catfish samples (2.0 × 2.0 × 2.0 cm) were put into the NMR tubes for determination. The parameters in moisture measurement were set as follows: SW = 100, SF = 20, NS = 4, TW = 1000 ms, TE = 1.00 ms, NECH = 1000, PRG = 1. 

### 2.10. ATP-Related Compounds 

Adenosine triphosphate (ATP)-related compounds were extracted according to Shao, et al. [25] and were analyzed by high-performance liquid chromatograph (HPLC) (Waters e2695 Separations Module and 2998 PDA Detector, USA) equipped with an SPD-10A (V) detector and a VP-ADS C18 column (4.6 mm i.d. × 250 mm, 5 μm particle size). The mobile phase was a phosphate buffer (pH 6.0) consisting of 0.02 mol/L K2HPO4 solution and 0.02 mol/L KH2PO4 solution (1: 1, *v*/*v*), and the flow rate was 1.0 mL/min. The samples were filtered through a 0.22 μm membrane, analysis was carried out by injecting 20 μL of sample, and detection was monitored at 254 nm. The temperature of the column oven was set at 35 °C and maintained. Each sample was run for 45 min. All reagents were of chromatographic grade.

### 2.11. Statistical Analysis

All data were expressed as mean ± standard deviation and analysis of variance. The measurements for each parameter were repeated three times. The statistical significance was identified at the 95% confidence level (*p* < 0.05) and was calculated by SPSS 26 (Chicago, IL, USA) software. Origin 2018 (Origin-Lab, Northampton, MA, USA) software was used for data processing and chart plotting.

## 3. Results and Discussion

### 3.1. Microbial Analysis

The changes in TVC values of catfish fillets during storage at 4 °C are shown in Figure 1A. The initial microbial counts of fresh catfish fillets were 4.88 ± 0.08 log CFU/g. A lag phase of 3 days was observed in all groups. From day 3 to 5, the TVC values of GSE, LRPE, AA and the CK group increased significantly (*p* < 0.05), with a nonsignificant increase (*p* > 0.05) observed in the LSPC group. After being stored at 4 °C for 5 days, an exponential augment occurred in the TVC value of each group. On day 7, the highest TVC value was observed in CK (8.58 ± 0.07 log CFU/g), followed by the LRPE, AA and GSE groups (7.62 ± 0.14, 7.48 ± 0.12 and 7.03 ± 0.06 log CFU/g, respectively), while the LSPC group had the lowest TVC value (6.71 ± 0.17 log CFU/g), indicating that GSE, LSPC, LRPE and AA could effectively inhibit the growth of microorganisms in catfish fillets during storage. The foremost among them was LSPC, which showed excellent antibacterial activity. Wang, Xie and Sun [10] believed that the antibacterial activity of LSPC was associated with the interaction of LSPC and the cell membrane of spoilage organisms. LSPC contains rich B-type procyanidins; among them, the hydrophobic domain in catechins could band with the surface choline group of the lipid membrane [27]. As a result, the cell membrane is damaged, resulting in the escape of cell contents. These stresses destroy the intracellular homeostasis environment, while other studies have reported that GSE revealed broadly antimicrobial activities against both Gram-positive and Gram-negative bacteria [28,29]. According to the International Commission on Microbiological Specifications for Foods, the upper acceptability limit of microorganisms in freshwater fish is 7.0 log CFU/g [30], implying that after 7 days of storage at 4 ℃, catfish fillets of all groups except for LSPC were rotten and unfit for consumption. At the end of storage, the bacterial counts in all samples exceeded 9 log CFU/g, which was far beyond the upper tolerable limit for fresh and refrigerated catfish fillets.

Furthermore, the microbiota composition of the initial (day 0), intermediate stage (day 7), and spoiled (day 11) catfish fillets were analyzed through high-throughput sequencing based on the Illumina MiSeq platform. After filtering the low-quality sequences, sequencing on the Illumina MiSeq platform resulted in a total of 1,730,008 effective sequences (Table 1). The Good’s coverage was ≥0.999, suggesting that almost all microbial phylotypes in catfish fillets were identified.

The composition and relative abundance of bacterial communities at the phylum level are shown in Figure 1B. The dominant bacteria in fresh samples of the five groups are *Proteobacteria*, which accounted for 80.64%, 46.21, 45.83%, 62.75%, and 66.33% in the CK, AA, GSE, LSPC, and LRPE groups, respectively. The addition of bacteriostatic substances significantly (*p* < 0.05) reduced the relative abundance of *Proteobacteria* on day 0. Among them, the antibacterial activity of GSE could be attributed to its phenolic acids, catechins and proanthocyanidins, which were confirmed to be the most potent chemicals against bacteria [31,32]. Other bacterial phyla, such as *Thermales*, *Firmicutes*, *Actinobacteria*, and *Bacteroidetes*, were identified in CK with a proportion of 12.62%, 4.90%, 1.57%, and 0.17%, respectively. The diversity of the microbiota community in refrigerated catfish fillets dropped drastically with time, and only a tiny portion of bacteria phylum engaged in the spoiling process. On days 7 and 11, the proportion of *Proteobacteria* in all groups reached more than 95%, followed by the *Firmicutes* of 0.33~4.20%. The remaining phyla were very low in abundance, including *Verrucomicrobia*, *Fusobacteria*, *Chloroflexi*, *Cyanobacteria*, *Acidobacteria*, *Gemmatimonadetes*, *Chlorobi*, *Deferribacteres* and *Tenericutes* (less than 0.3% of total sequences).

The composition and relative abundance of different genera are shown in Figure 1C. At the genus level, the dominant strains of fresh catfish fillets are *Pseudomonas*, *Burkholderia*, *Thermus*, and *Anoxybacillus*, accounting for 14.21%, 61.13%, 12.67%, and 2.19%, respectively, in the CK group. The addition of bacteriostatic substances significantly reduced the relative abundance of *Pseudomonas* at the beginning of storage (*p* < 0.05). Li, et al. [8] also observed that *Pseudomonas* showed an increasing trend during the 17-day storage period of refrigerated beef, and the addition of LSPC could retard the reproduction of *Pseudomonas*. However, in the later stage of storage (day 11), the relative abundance of *Pseudomonas* in the CK, AA, GSE, LSPC, and LRPE groups grew significantly to 97.25%, 80.87%, 83.04%, 88.77%, and 82.90%, respectively. 

A heatmap was used to exhibit the microbiota differences between the five groups. According to Figure 1D, the color reflects the relative abundances of the microbiological genera, with the redder and the bluer colors illustrating the higher and the lower relative abundances, respectively. The cluster trees on the left and top were added based on the similarity of genera abundances. The most diversified bacterial composition was observed in the fresh sample. As previous research described, the *Aeromonas*, *Acinetobacter*, *Moraxella*, *Pseudomonas*, *Shewanella* and other gramnegative bacteria were usually contained in freshwater fish from temperate waters [33,34]. In the present study, *Burkholderia* and *Thermus* were predominant in the samples stored on day 0. Despite that the genus richness of each group decreased sharply with increasing storage time, the genus richness of the spoiled CK (CK_11d) was still substantially higher than the other groups. The relative abundance of *Pseudomonas* in CK sharply increased from 14.21% (day 0) to 62.88% (day 11). 

### 3.2. TVB-N and TBARS

The changes in TVB-N values of catfish fillets during storage at 4 °C are displayed in Figure 2A. After 3 days, the TVB-N values of each group changed slightly, which was similar to the trends of TVC. From day 3 to 7, the TVB-N value of CK increased and was higher than in other groups (*p* < 0.05). Subsequently, the TVB-N value of each group increased dramatically, and the TVB-N value of CK exceeded the limit level on day 9 (27.55 ± 0.96 mg/100 g), considered as serious spoilage. The rapid augment of the TVB-N value occurred in the late storage period, which can be attributed to the essential long degradation process for the generation of TVB-N from convent nitrogen-containing macromolecules to volatile small molecular compounds under the action of microbes [35]. Liu, et al. [36] and Yu, et al. [37] believe that the sharp increase in TVB-N in the later stages of storage was also associated with the increase in pH. At the end of storage (day 11), the TVB-N value of the GSE, LSPC, LRPE and AA groups was 27.99 ± 1.01, 28.05 ± 1.02, 29.29 ± 1.06 and 22.91 ± 0.84 mg/100 g, respectively, all of which were significantly lower (*p* < 0.05) than CK (37.75 ± 1.34 mg/100 g), indicating that the GSE, LSPC, LRPE groups had a similar effect on reducing TVB-N production. Furthermore, in later storage, the effect of AA on lowering TVB-N formation was better than that of GSE, LSPC and LRPE, most likely because AA, as a small molecule substance, was more easily absorbed by the catfish fish matrix and played an antioxidant role for a long time. In summary, GSE, LSPC and LRPE are conducive in reducing TVB-N production, thereby extending the storage time of catfish samples.

In the present study, TBARS values of different samples during storage are shown in Figure 2B. The results suggest that TBA values of the five groups increased continuously from the initial 0.12 ± 0.03 mg MDA/kg to 0.73 ± 0.02, 0.61 ± 0.02, 0.52 ± 0.00, 0.58 ± 0.02, and 0.59 ± 0.09 mg MDA/kg after 11 days of storage for the CK, AA, GSE, LSPC and LRPE group, respectively, indicating that lipid oxidation occurred during the whole storage process. According to Fan, et al. [38], the increase in TBARS values during refrigerated storage could be attributed to the partial dehydration of fish as well as to the enhanced oxidation of unsaturated fatty acids. Different from TVB-N, a one day of lag phase was observed in TBARS values of all groups except for CK in the early stage of storage. As for CK, the TBARS value was significantly higher than that of other groups and increased rapidly throughout storage (*p* < 0.05), further confirming the anti-lipid oxidation activity of GSE, LSPC, LRPE and AA. Polyphenols, according to Li, et al. [26], may play a role in protecting the endogenous enzyme antioxidant system, thus blocking the chain reaction of lipid oxidation. Among the four groups, the GSE-treated group had the best effect. It was reported that the GSE demonstrated high anti-lipid oxidant activity in rainbow trout meat [39]. It was probably because that GSE contained 98% total flavanols, including 89% proanthocyanidins, which had high antioxidant activity [40]. Because of the antioxidant properties, polyphenol-rich plant extracts such as GSE could be widely used in food processing and preservation [28]. 

### 3.3. Shear Force and Histology

The initial shear force of fresh catfish fillets was 2287 ± 49 g, as shown in Figure 2C, and a steady lower trend was seen in all groups during storage at 4 °C. It was obvious that the changes in shear force of each group, except for LRPE and LSPC, could be roughly divided into two stages, namely, the sharp decline in the first three days at the early storage period and the slow decline from the third day to the end of storage. As for the LRPE and LSPC groups, the texture deterioration was slower than in the other groups. Meanwhile, the shear force of the LRPE group remained the highest from days 3–11, indicating that LRPE effectively inhibited texture deterioration of the catfish fillets. 

The changes of the microstructure in catfish fillets during storage are depicted in Figure 3. It was obvious that, as storage time progressed, cracks appeared in the muscle fibers of each group, and the fiber gaps widened. At the initial stage of storage, the sarcomeres of the catfish fillets treated with GSE, LSPC, and LRPE were closely arranged, and no obvious damage and breakage were found, while that of the CK and AA group were somewhat fractured, with some space between the tissues. On day 7, the integrity of the cell morphologies was better in the AA group than in the GSE, LSPC and LRPE groups, demonstrating that the addition of AA may alleviate tissue structure deterioration in catfish fillets better than in GSE, LSPC and LRPE during the middle storage period. After 11 days of storage at 4 °C, the samples in each group revealed varying degrees of fiber damage, among which CK had the most extensive damage, with cells dispersed and vacuolated, whereas the GSE, LSPC, LRPE, and AA groups displayed relatively intact histomorphology. The microstructure alterations of meat and meat products are intimately connected to their quality features, particularly water-holding capacity [41]. The retention of water in catfish fillets by polyphenols may be one of the reasons for delayed muscle fiber damage, which corresponds to the results of centrifugal loss (Figure 2D). 

### 3.4. Centrifugal Loss, Water Molecules Distribution, and MRI Analysis

As shown in Figure 2D, the reduction of WHC could be ascribed to the denaturation and aggregation of myofibrillar protein in fish during cold storage, which led to the increase in centrifugation loss during the first 9 days storage. The centrifugal loss decreased in the last 2 days of the storage period, which may be attributed to the large amount of loss of water in the first 9 days, resulting in less free water in the catfish fillets. During the storage period, the centrifugal loss of the CK was significantly higher than that of the other groups (*p* < 0.05), indicating that GSE, LSPC, LRPE and AA could effectively inhibit water loss in the catfish fillets. GSE produced the greatest effects of all, which was probably because the proanthocyanidins in GSE have a significant effect in delaying myofibrillar protein denaturation and aggregation.

Three water components were identified according to different water activities, referred to as T_2b_ (0–10 ms), T_21_ (10–100 ms), and T_22_ (>100 ms), which represented bound water, immobilized water and free water, respectively (Table 2). In general, T_2b_ does not alter much during storage because the bound water does not relate to changes in mechanical stress and micro- or macrostructure in the meat matrix and is usually tightly attached to muscle protein. However, T_2b_ increased with storage time in the present research, which could be attributed to discrepancies in the individual catfish and experimental settings. With the extension of storage time, a transformation from long relaxation time to short relaxation time was observed in T_21_, and the most pronounced trend was seen in the CK, indicating that the water was more and more active and the binding capacity of muscle tissue to water molecules was becoming weaker and weaker, which could be conjectured as the hydrolysis of the catfish fillet muscle proteins resulting from bacteria or enzymes [42]. With the addition of polyphenols, the process of protein oxidative denaturation was delayed, creating a binding between protein and water molecules that was more difficult to destroy. As a result, the T_21_ of the GSE, LSPC, and LRPE groups moved from a short to long relaxation time more slowly than the CK. Moreover, the structures of these proteins can be changed by lactic acid fermentation of meat [43]. The increase in T_22_ could be attributed to the dissociation of immobilized water [44], which could be confirmed from the reduction of P_21_ in Table 3. As shown in Table 3, at any given period, the combination of P_21_ and P_22_ accounted for more than 95% of the total area, suggesting that the majority of water was present in catfish fillets as free water and immobilized water [43]. 

MRI was used to visualize the internal structure of the catfish fillets during refrigerated storage in a lossless way (Figure 4). In the MRI, red represents high proton density regions and green represents low proton density regions. The stronger the water proton signal, that is, the redder the image color, the higher the water content. As shown in Figure 4, the MRI of all samples at day 0 revealed yellow red, suggesting that the water content of catfish fillets was high at the beginning of storage. As storage time progressed, the MRI of catfish fillets altered from red to green, suggesting that the moisture content steadily reduced. The hue of the CK was the greenest at the end of storage, indicating that its water loss was the most significant and that the water loss could be inhibited by polyphenols. 

### 3.5. ATP-Related Compounds

In general, ATP is degraded by autolytic breakdown, endogenous enzymes, as well as by bacterial activity, including *Pseudomonas* spp., *S. putrefaciens*, and *P*. *phosphoreum* and follows the following process: adenosine triphosphate (ATP) → adenosine diphosphate (ADP) → adenosine monophosphate (AMP) → inosine monophosphate (IMP) → and hypoxanthine riboside (HxR) → hypoxanthine (Hx) → xanthine → uric acid [45,46]. The changes in ATP-related compounds and K values are depicted in Figure 5. Regarding the changes in ATP-related compounds, the levels of IMP, HxR, and Hx varied substantially. The contents of ATP, ADP, and AMP showed a significant downward trend in the early storage period and then slowly decreased. The rapid decrease in the three ATP-related compounds during the early storage period may correspond to the expeditious degradation of ATP. The levels of ATP, ADP, and AMP in the CK were reduced by 89.95%, 48.22%, and 97.49%, respectively, at the end of storage, further confirming the degradation process of ATP. The most significant declining trend was noticed in the CK, indicating that GSE, LSPC, LRPE, and AA delayed the degradation of ATP to a certain extent.

As shown in Figure 5C, the initial level of IMP was 6.77 ± 0.25 μmol/g, which was significantly (*p* < 0.05) higher than the concentration of other ATP-related compounds at the early stage of storage. The IMP content in all groups demonstrated a progressive decreasing tendency with the passage of time. The IMP value in the LSPC group was significantly (*p* < 0.05) higher than that of the other groups during the middle storage period (day 1 to 7), especially on the third day of storage, whenever the IMP value of the LSPC group was nearly twice (1.89 times) as much as that of the CK, indicating that LSPC could delay ATP degradation and IMP accumulation to a considerable extent. The IMP value of each group remained steady after storage for 9 days, with no significant (*p* > 0.05) change. Additionally, GSE, Zhao, et al. [47] discovered a similar result when storing tilapia (*Oreochromis niloticus*) fillets, namely that the IMP content of GSE group was greater than that of the CK, of which it was possible that IMP degradation was restrained by GSE addition. Li, Zhuang, Liu, Zhang, Liu, Cheng, Liu, Shu and Luo [31] further explained that it was probably because of the inhibitory effect of GSE on the related enzymes activities. 

As displayed in Figure 5D, the level of HxR in all groups increased at the early stage and then declined. The HxR content of the CK increased from 0.05 ± 0.01 to 2.71 ± 0.05 μmol/g during the first 9 days of storage and decreased to 2.23 ± 0.01 μmol/g on day 11. Compared with CK, the three polyphenol extracts and AA treatment prevented the accumulation of HxR by 0.93~1.20 μmol/g at the end of storage. Along with the accumulation of HxR, the content of Hx in the fillets also increased, and the discrepancy between the groups became increasingly noticeable. On day 11, the Hx levels in the GSE, LSPC, LRPE, and AA groups were 1.98 ± 0.28, 2.41 ± 0.07, 2.79 ± 0.47 and 3.31 ± 0.41 μmol/g, respectively, considerably (*p* < 0.05) lower than that of the CK (3.77 ± 0.07 μmol/g). Given that the breakdown process from IMP to Hx was significantly influenced by spoilage bacteria, one possible explanation for the decreased levels of Hx in the GSE, LSPC and LRPE groups might be due to the polyphenols’ antibacterial activity [47]. It could be deduced that the reduced accumulation of HxR and Hx in the GSE-, LSPC-, LRPE- and AA-treated groups will result in a beneficial effect on fillet edibility via increasing flavor quality after refrigeration.

As depicted in Figure 6, the initial K value was far below 10%, suggesting that the catfish fillets were quite fresh. For CK, the K value increased steadily, reaching 81.70% on day 9 before decreasing slightly (*p* > 0.05). At the end of storage (day 11), the K value of the CK was approximately 19.2~25.6% higher than that of the other groups, demonstrating the capacities of GSE, LSPC, LRPE, and AA to inhibit the degradation of ATP-related compounds. The LSPC group had the strongest impact in suppressing the increase in K value, especially on day 7, which was 40.79% less than the CK. This might be attributable to the excellent bacteriostatic action of LSPC, which has been confirmed by Li, et al. [8].

## 4. Conclusions

GSE, LSPC, LRPE, and AA in this study showed strong effects on delaying the quality deterioration of catfish fillets during storage at 4 °C. Among them, GSE showed a good effect in inhibiting fat oxidation and water loss, while LSPC represented great antibacterial activity. Additionally, the effect of AA on lowering TVB-N formation was better than that of GSE, LSPC and LRPE, while LRPE had a strong ability to delay the decline of catfish shear force. These results provide a theoretical basis for the application of GSE, LSPC, LRPE, and AA as natural preservative in meat preservation. 

## Figures and Tables

**Figure 1 foods-12-00765-f001:**
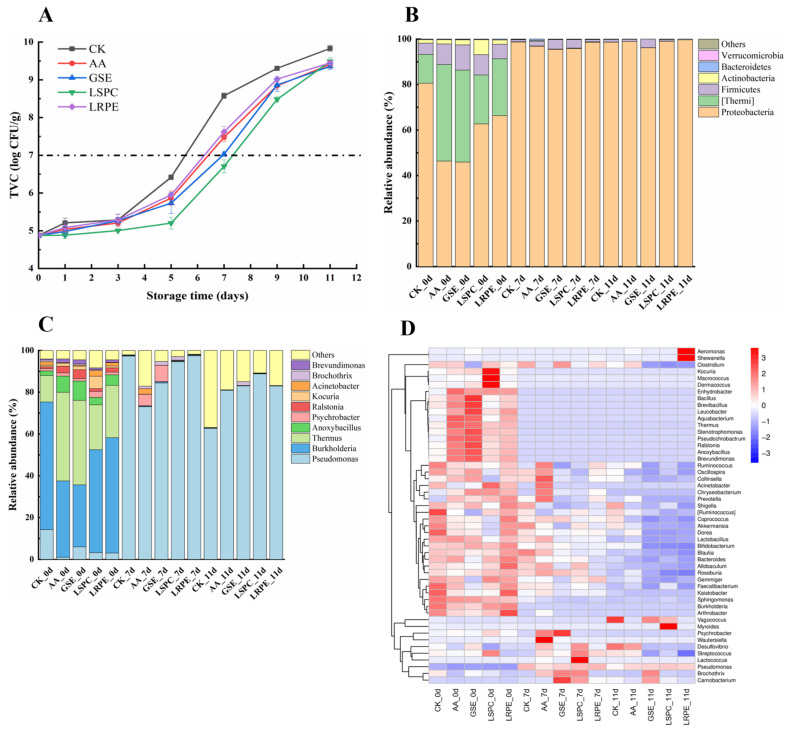
Changes in TVC (**A**), relative abundance of bacterial at phylum (**B**) and genus (**C**) level and heatmap (**D**) at genus level based on 16S rDNA sequencing. The category “others” represents a collection of genus with relative abundance less than 1%. (0 d: samples on day 0; 7 d: samples on day 7; 11 d: samples on day 11.).

**Figure 2 foods-12-00765-f002:**
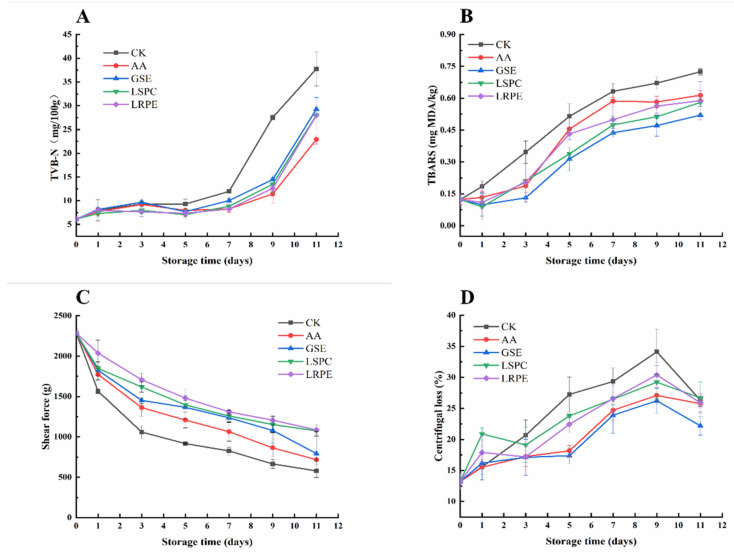
Changes in TVB-N (**A**), TBARS (**B**), centrifugal loss (**C**), and shear force (**D**) of CK-, GSE-, LSPC-, LRPE- and AA-treated catfish fillets during storage at 4 °C for the indicated times.

**Figure 3 foods-12-00765-f003:**
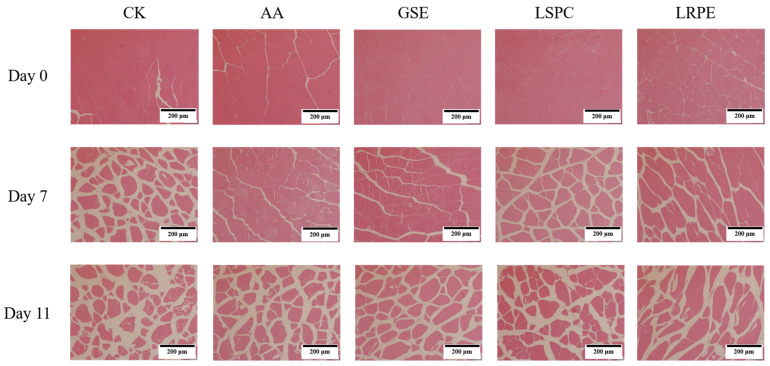
Changes in microstructure profiles of CK-, GSE-, LSPC-, LRPE- and AA-treated catfish fillets during storage at 4 °C for the indicated times.

**Figure 4 foods-12-00765-f004:**
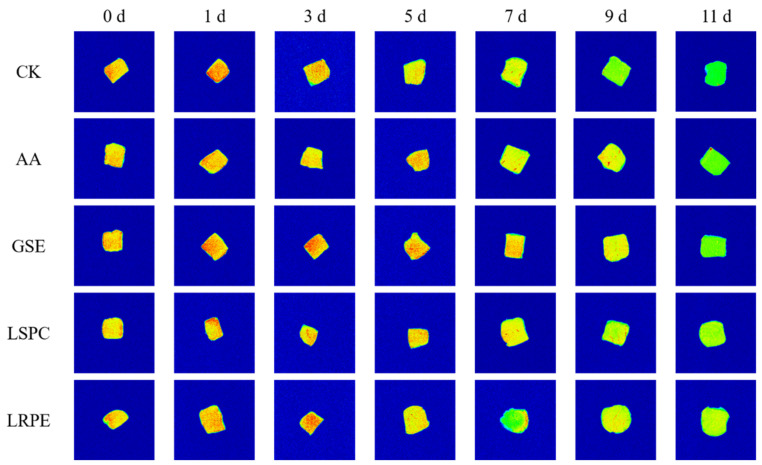
Changes in false-color image of the water proton density of CK-, GSE-, LSPC-, LRPE- and AA-treated catfish fillets during storage at 4 °C for the indicated times.

**Figure 5 foods-12-00765-f005:**
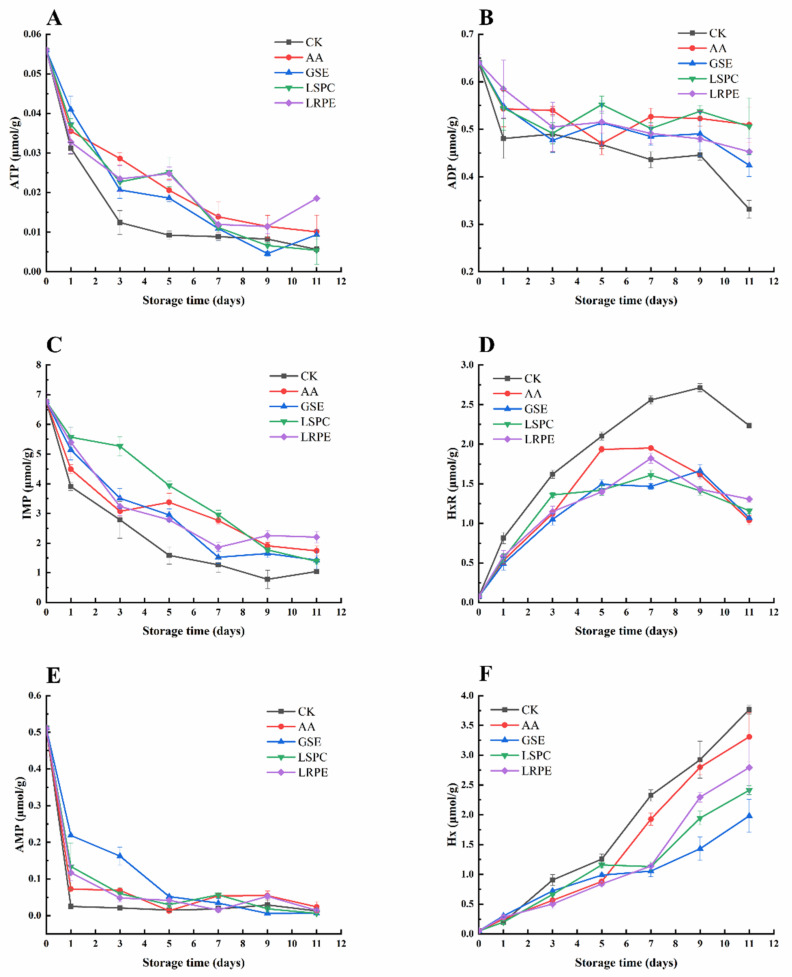
Changes in ATP concentration (**A**), ADP concentration (**B**), IMP concentration (**C**), HxR concentration (**D**), AMP concentration (**E**), and Hx concentration (**F**) of CK-, GSE-, LSPC-, LRPE- and AA-treated catfish fillets during storage at 4 °C for the indicated times.

**Figure 6 foods-12-00765-f006:**
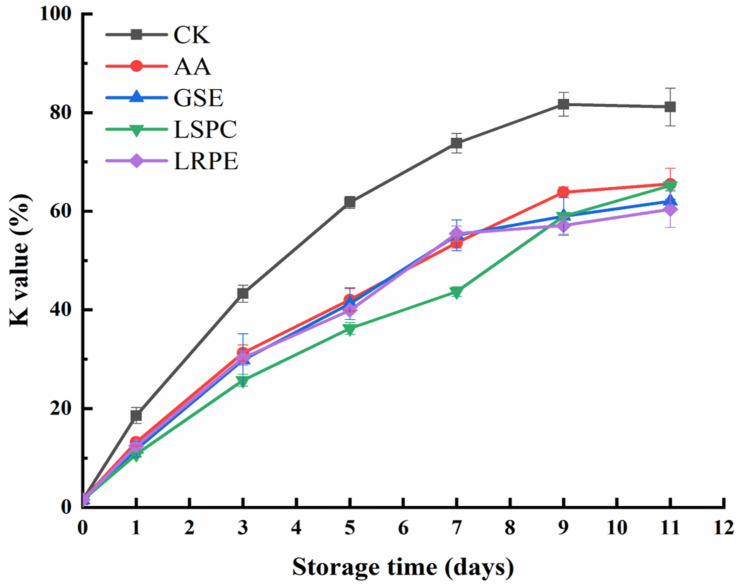
Changes in K value of CK-, GSE-, LSPC-, LRPE- and AA-treated catfish fillets during storage at 4 °C for the indicated times.

**Table 1 foods-12-00765-t001:** Alpha diversity estimation of bacterial communities in CK-, GSE-, LSPC-, LRPE- and AA-treated catfish fillets during storage at 4 °C for the indicated times.

Samples	Sequences	OTUs	Chao	Good’s Coverage	Shannon	Simpson
CK-0 d	96,878	287	374	0.9993	2.81	0.73
AA-0 d	105,787	252	350	0.9990	2.92	0.68
GSE-0 d	110,338	275	371	0.9992	3.25	0.77
LSPC-0 d	117,002	256	391	0.9991	3.14	0.74
LRPE-0 d	96,209	267	377	0.9991	2.60	0.63
CK-7 d	112,132	185	289	0.9991	3.35	0.83
AA-7 d	104,106	192	294	0.9992	3.49	0.83
GSE-7 d	126,224	169	299	0.9993	3.46	0.79
LSPC-7 d	129,438	176	312	0.9991	2.83	0.64
LRPE-7 d	116,472	187	234	0.9994	1.53	0.33
CK-11 d	127,863	165	250	0.9995	2.73	0.69
AA-11 d	128,558	151	237	0.9994	2.30	0.52
GSE-11 d	124,207	98	152	0.9997	1.76	0.41
LSPC-11 d	107,973	104	132	0.9998	1.81	0.42
LRPE-11 d	126,801	83	135	0.9998	1.93	0.45

**Table 2 foods-12-00765-t002:** Changes in relaxation time of CK-, GSE-, LSPC-, LRPE- and AA-treated catfish fillets during storage at 4 °C for the indicated times. Different capital letters at the same day of storage are significantly different (*p* < 0.05) between different groups. Different lowercase letters at the same group are significantly different (*p* < 0.05) between different storage time.

Relaxation Time	Samples	Storage Time (Days)						
		0	1	3	5	7	9	11
T_21_ (ms)	CK	0.89 ± 0.08 ^Ad^	1.02 ± 0.11 ^Ad^	1.26 ± 0.09 ^Ac^	1.31 ± 0.06 ^Ac^	1.38 ± 0.08 ^Abc^	1.51 ± 0.11 ^Ab^	2.25 ± 0.14 ^Aa^
	AA	0.89 ± 0.08 ^Ae^	1.09 ± 0.08 ^Acd^	0.98 ± 0.02 ^Bde^	1.17 ± 0.12 ^Abc^	1.24 ± 0.08 ^Aabc^	1.28 ± 0.11 ^Cab^	1.34 ± 0.10 ^Ba^
	GSE	0.89 ± 0.08 ^Ac^	1.13 ± 0.13 ^Ab^	1.20 ± 0.04 ^Aab^	1.18 ± 0.12 ^Aab^	1.29 ± 0.07 ^Aab^	1.27 ± 0.06 ^Cab^	1.34 ± 0.05 ^Ba^
	LSPC	0.89 ± 0.08 ^Ac^	1.04 ± 0.13 ^Ac^	1.08 ± 0.04 ^Bbc^	1.11 ± 0.19 ^Abc^	1.30 ± 0.19 ^Aab^	1.74 ± 0.04 ^ABa^	1.73 ± 0.03 ^Ba^
	LRPE	0.89 ± 0.08 ^Ac^	1.05 ± 0.10 ^Ab^	1.31 ± 0.07 ^Aa^	1.32 ± 0.12 ^Aa^	1.36 ± 0.08 ^Aa^	1.32 ± 0.04 ^BCa^	1.42 ± 0.10 ^Ba^
T_22_ (ms)	CK	53.54 ± 0.05 ^Ad^	61.10 ± 0.33 ^Ac^	63.29 ± 5.30 ^Abc^	63.29 ± 4.52 ^Abc^	64.87 ± 0.41 ^Abc^	66.46 ± 2.04 ^Ab^	78.21 ± 0.08 ^Aa^
	AA	53.54 ± 0.05 ^Ab^	61.20 ± 0.33 ^Aa^	61.57 ± 2.22 ^Aa^	57.17 ± 2.60 ^Bab^	63.16 ± 6.34 ^Aa^	63.13 ± 2.59 ^Ba^	63.17 ± 4.11 ^Ba^
	GSE	53.54 ± 0.05 ^Ac^	60.61 ± 0.41 ^Ab^	60.31 ± 0.37 ^Ab^	63.50 ± 2.68 ^Aab^	63.95 ± 3.28 ^Aa^	61.46 ± 1.00 ^Bab^	60.51 ± 0.73 ^Bb^
	LSPC	53.54 ± 0.05 ^Ad^	56.25 ± 1.13 ^Bcd^	63.16 ± 0.90 ^Aab^	59.27 ± 1.41 ^ABbc^	60.42 ± 3.99 ^Abc^	63.34 ± 0.47 ^Bab^	66.48 ± 4.93 ^Ba^
	LRPE	53.54 ± 0.05 ^Ad^	55.86 ± 2.07 ^Bcd^	60.11 ± 0.89 ^Aab^	57.24 ± 2.26 ^Bbc^	63.13 ± 1.87 ^Aa^	62.84 ± 1.10 ^Ba^	62.84 ± 3.22 ^Ba^
T_23_ (ms)	CK	550.73 ± 12.14 ^Ae^	691.73 ± 2.01 ^Ad^	695.99 ± 19.12 ^Ad^	709.62 ± 12.95 ^Ad^	815.47 ± 25.18 ^Ac^	926.52 ± 21.49 ^Ab^	1259.95 ± 21.71 ^Aa^
	AA	550.73 ± 12.14 ^Ae^	635.71 ± 17.08 ^Bd^	654.40 ± 8.66 ^Bd^	699.58 ± 6.25 ^Ac^	718.55 ± 16.19 ^Bc^	827.21 ± 14.68 ^Bb^	860.71 ± 10.86 ^Ba^
	GSE	550.73 ± 12.14 ^Ad^	592.51 ± 6.66 ^Cc^	610.87 ± 20.60 ^Cbc^	618.77 ± 8.97 ^Bb^	629.26 ± 13.57 ^Db^	724.20 ± 16.21 ^Ca^	740.49 ± 5.78 ^CDa^
	LSPC	550.73 ± 12.14 ^Ae^	551.27 ± 17.50 ^De^	582.79 ± 18.47 ^Cd^	628.44 ± 9.16 ^Bc^	654.74 ± 2.88 ^CDb^	705.71 ± 12.19 ^CDa^	726.23 ± 15.47 ^Da^
	LRPE	550.73 ± 12.14 ^Af^	595.52 ± 5.68 ^Ce^	607.93 ± 6.30 ^Cde^	617.33 ± 3.70 ^Bd^	662.58 ± 15.62 ^Cc^	683.92 ± 3.34 ^Db^	757.46 ± 17.12 ^Ca^

**Table 3 foods-12-00765-t003:** Changes in water distribution of CK-, GSE-, LSPC-, LRPE- and AA-treated catfish fillets during storage at 4 ℃ for the indicated times. Different capital letters at the same day of storage are significantly different (*p* < 0.05) between different groups. Different lowercase letters at the same group are significantly different (*p* < 0.05) between different storage time.

Water Distribution	Samples	Storage Time (Days)						
		0	1	3	5	7	9	11
P_21_ (%)	CK	1.01 ± 0.44 ^Ac^	1.65 ± 0.26 ^BCc^	2.74 ± 0.48 ^Ab^	2.76 ± 0.29 ^Ab^	2.52 ± 0.37 ^Ab^	3.04 ± 0.25 ^Aab^	3.60 ± 0.72 ^Aa^
	AA	1.01 ± 0.44 ^Ac^	1.00 ± 0.25 ^Cc^	1.93 ± 0.16 ^Ab^	2.15 ± 0.86 ^Aab^	2.64 ± 0.42 ^Aab^	2.86 ± 0.20 ^Aa^	2.78 ± 0.25 ^ABa^
	GSE	1.01 ± 0.44 ^Ac^	1.84 ± 0.79 ^Bbc^	2.30 ± 0.77 ^Aab^	2.37 ± 0.34 ^Aab^	2.52 ± 0.21 ^Aab^	2.87 ± 0.26 ^Aa^	2.47 ± 0.36 ^Bab^
	LSPC	1.01 ± 0.44 ^Ac^	1.61 ± 0.09 ^BCbc^	1.99 ± 0.66 ^Aabc^	3.19 ± 1.64 ^Aa^	3.02 ± 0.89 ^Aa^	2.83 ± 0.42 ^Aab^	2.56 ± 0.32 ^Bab^
	LRPE	1.01 ± 0.44 ^Ab^	2.67 ± 0.27 ^Aa^	2.67 ± 0.23 ^Aa^	2.97 ± 0.64 ^Aa^	2.60 ± 0.34 ^Aa^	2.50 ± 0.49 ^Aa^	1.63 ± 0.48 ^Cb^
P_22_ (%)	CK	98.70 ± 0.47 ^Aa^	97.94 ± 0.06 ^ABa^	96.82 ± 0.68 ^Ab^	96.66 ± 0.56 ^Ab^	95.95 ± 0.29 ^Cbc^	96.03 ± 0.20 ^Abc^	95.29 ± 0.72 ^Bc^
	AA	98.70 ± 0.47 ^Aa^	98.74 ± 0.33 ^Aa^	97.95 ± 0.21 ^Aa^	97.13 ± 0.73 ^Ab^	96.96 ± 0.61 ^ABb^	96.72 ± 0.16 ^Abc^	96.02 ± 0.35 ^ABc^
	GSE	98.70 ± 0.47 ^Aa^	97.49 ± 1.15 ^Bb^	97.00 ± 0.61 ^Abc^	96.58 ± 0.39 ^Abcd^	96.53 ± 0.45 ^ABCbcd^	96.22 ± 0.49 ^Acd^	95.54 ± 0.50 ^ABd^
	LSPC	98.70 ± 0.47 ^Aa^	97.92 ± 0.21 ^ABab^	97.07 ± 1.08 ^Abc^	96.21 ± 1.41 ^Ac^	96.03 ± 0.70 ^BCc^	96.40 ± 0.43 ^Ac^	96.49 ± 0.43 ^Ac^
	LRPE	98.70 ± 0.47 ^Aa^	96.94 ± 0.19 ^Bb^	97.09 ± 0.23 ^Ab^	96.13 ± 0.66 ^Ab^	97.07 ± 0.34 ^Ab^	96.47 ± 0.84 ^Ab^	96.37 ± 0.38 ^Ab^
P_23_ (%)	CK	0.28 ± 0.08 ^Ad^	0.40 ± 0.29 ^Acd^	0.44 ± 0.20 ^ABcd^	0.58 ± 0.28 ^Abcd^	1.53 ± 0.63 ^Aa^	0.96 ± 0.01 ^Abc^	1.11 ± 0.04 ^Bab^
	AA	0.28 ± 0.08 ^Acd^	0.27 ± 0.10 ^Acd^	0.12 ± 0.12 ^Bd^	0.72 ± 0.15 ^Ab^	0.40 ± 0.24 ^Bc^	0.42 ± 0.06 ^Bc^	1.20 ± 0.10 ^Ba^
	GSE	0.28 ± 0.08 ^Ac^	0.33 ± 0.11 ^Ac^	0.91 ± 0.52 ^Ab^	1.06 ± 0.38 ^Ab^	0.95 ± 0.28 ^ABb^	0.90 ± 0.27 ^ABb^	1.99 ± 0.18 ^Aa^
	LSPC	0.28 ± 0.08 ^Ac^	0.50 ± 0.09 ^Aabc^	0.94 ± 0.46 ^Aa^	0.46 ± 0.03 ^Abc^	0.95 ± 0.22 ^ABa^	0.77 ± 0.01 ^ABab^	0.96 ± 0.28 ^Ba^
	LRPE	0.28 ± 0.08 ^Ac^	0.39 ± 0.08 ^Ac^	0.24 ± 0.01 ^Bc^	0.90 ± 0.49 ^Ab^	0.31 ± 0.03 ^Bc^	1.03 ± 0.53 ^Ab^	2.00 ± 0.16 ^Aa^

## Data Availability

The authors confirm that all data included in this study are available upon request by contacting the corresponding author.

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
