# Peer review of "Three Phenolic Extracts Regulate the Physicochemical Properties and Microbial Community of Refrigerated Channel Catfish Fillets during Storage"

_foods, 2023, doi:10.3390/foods12040765_

Round 1

Reviewer 1 Report

The research and the methodology are interesting, however there is some aspects to improve.

Title: the research evaluates the effectivity of phenolic compounds mixes over catfish fillets deterioration, it is suggested adequate the title. Likewise, it is suggested adjusting the remaining of the document, in example conclusion on line 97 that reference to three polyphenols which will be three mixes of polyphenols.

Line 20: What means Good effect?, could you provide or explain with data

Line 45: use the same style for references citation, in example delete L de “L. Shi, et al., 2020”

Line 152. Include a brief description over TBARS determination.  

Reviewer 2 Report

This MS summarizes the effects of various polyphenols (grape seed (GSE), lotus seedpod (LSPC), and lotus root (LRPE)) on changes in viable bacterial counts, bacterial flora, and physicochemical properties to prolong catfish fillets storability. The storability of catfish fillets has been evaluated from multiple perspectives and the quality of the data is high. However, there are a few points of concern.

Comments

(1)     Throughout: Three extracts (grape seed (GSE), lotus seedpod (LSPC), and lotus root (LRPE)) are used as polyphenol sources. The authors attempt to explain the differences in effectiveness by the differences in the chemical structure of the polyphenols contained in them (GSE, LSPC, LRPE). However, there is no data on the total polyphenol content in the extracts (GSE, LSPC), which should be measured. In addition, the polyphenol content taken up by the fillets is likely to differ among the different polyphenol types (GSE, LSPC, LRPE). Therefore, the total polyphenol content in the fillets should also be measured. With this data, I would be able to make a deeper study on the effect of polyphenols on maintaining the freshness of fillets.

(2)     Data: All tables and graphs do not show statistical processing results. The authors need to show the statistical results in tables and graphs as well as in text.

(3)     L20-21: This sentence contains a conclusion. Isn't this sentence unnecessary?

(4)     L21-22: I don't know what increase of catfish fillets was inhibited.

(5)     L110-114: A more detailed method needs to be shown for the immersion conditions of the polyphenol solution. For example, temperature, time, agitation, etc.

(6)     Table1: How much is the n-number? Could it be that n=1? If so, I don't think the results of the bacterial flora can be published as a research paper.

(7)     Figure 1: There is a difference in the composition of the bacteria before the test between the test groups. The authors explain that this is due to the fact that each polyphenol exhibited antimicrobial activity. Since the analysis of the bacterial flora uses DNA, even dead bacteria can be detected, as opposed to live bacteria counts. Therefore, this consideration is not plausible.

Enterobacteriaceae is a family, not a genus.

(8)     Figure 3: Indicates magnification.

(9)     Figure 4: I don't know which color indicates which amount of moisture.

(10)  L252: I think 0.99% is an error. Looking at the graph, it seems to be about 14%.

(11)  L266: Not green, but blue?

(12)  L301-303: This sentence is wrong.

Reviewer 3 Report

The manuscript Foods-2159879 entitled “Three varieties of polyphenols regulate the physicochemical properties and microbial community of refrigerated channel catfish fillets during storage” investigated the ability of three different phenolic extracts (rape see, lotus seedpod, and lotus root) to extend the shelf life of refrigerated channel catfish fillets during storage at 4 °C. To achieve the goals, the authors tested the physicochemical changes and bacterial community. The objective and design of this study is clear and was satisfactorily carried out. The paper is written very nicely with precise language and format. The authors need to add the funding organization. Otherwise, I suggest the publication of this manuscript.

Round 2

Reviewer 2 Report

I am satisfied with the revisions that have been made by the authors.